# The Other Side of the Same Coin: Beyond the Coding Region in Amyotrophic Lateral Sclerosis

**DOI:** 10.3390/ph18101573

**Published:** 2025-10-18

**Authors:** Paola Ruffo, Benedetta Perrone, Francesco Perrone, Francesca De Amicis, Rodolfo Iuliano, Cecilia Bucci, Angela Messina, Francesca Luisa Conforti

**Affiliations:** 1Laboratory of Medical Genetics, Department of Pharmacy and Health and Nutritional Sciences, University of Calabria, 87036 Rende, Italy; paola.ruffo@unical.it (P.R.);; 2Health Center, Department of Pharmacy and Health and Nutritional Sciences, University of Calabria, 87036 Rende, Italy; 3Department of Health Sciences, Magna Græcia University of Catanzaro, 88100 Catanzaro, Italy; iuliano@unicz.it; 4Department of Experimental Medicine, University of Salento, 73100 Lecce, Italy; 5Department of Biological, Geological and Environmental Sciences, University of Catania, 95129 Catania, Italy

**Keywords:** transposable elements, amyotrophic lateral sclerosis, epigenetic regulation, retrotransposons

## Abstract

Transposable elements (TEs), once regarded as genomic “junk,” are now recognized as powerful regulators of gene expression, genome stability, and innate immunity. In the context of neurodegeneration, particularly Amyotrophic Lateral Sclerosis (ALS), accumulating evidence implicates TEs as active contributors to disease pathogenesis. ALS is a fatal motor neuron disease with both sporadic and familial forms, linked to genetic, epigenetic, and environmental factors. While coding mutations explain a subset of cases, advances in long-read sequencing and epigenomic profiling have unveiled the profound influence of non-coding regions—especially retrotransposons such as LINE-1, Alu, and SVA—on ALS onset and progression. TEs may act through multiple mechanisms: generating somatic mutations, disrupting chromatin architecture, modulating transcriptional networks, and triggering sterile inflammation via innate immune pathways like cGAS-STING. Their activity is normally repressed by epigenetic regulators, including DNA methylation, histone modifications, and RNA interference pathways; however, these controls are compromised in ALS. Taken together, these insights underscore the translational potential of targeting transposable elements in ALS, both as a source of novel biomarkers for patient stratification and disease monitoring, and as therapeutic targets whose modulation may slow neurodegeneration and inflammation. This review synthesizes the current knowledge of TE biology in ALS; integrates findings across molecular, cellular, and systems levels; and explores the therapeutic potential of targeting TEs as modulators of neurodegeneration.

## 1. Introduction

Neurodegenerative diseases represent a heterogeneous group of disorders characterized by the progressive and selective loss of neurons in the central nervous system (CNS), leading to distinct impairments in functional systems that define their varied clinical presentations. These debilitating conditions, including Alzheimer’s disease (AD), Parkinson’s disease (PD), Huntington’s disease (HD), and Amyotrophic Lateral Sclerosis (ALS), collectively pose immense global health challenges due to their devastating impact on patients’ quality of life and the substantial burden on healthcare systems. While AD is primarily associated with cognitive decline, PD with progressive motor deficits, and HD with chorea, ALS stands out as one of the most common and aggressive adult-onset neuromuscular diseases. It is characterized by the relentless degeneration of both upper and lower motor neurons in the brain and spinal cord, respectively, leading to progressive muscle weakness, paralysis, and ultimately death, typically from respiratory failure within 2–5 years of diagnosis [1,2].

The epidemiology of ALS shows notable geographic variability. The incidence in Europe and North America is typically reported at ~2–3 per 100,000 person-years, with the highest prevalence documented in Northern Europe (up to 9 per 100,000) [3]. In contrast, reported prevalence in Asia, including Japan and China, is lower (~1–2 per 100,000), while recent studies in Africa and South America remain limited, though suggest incidence values below those in Western countries [4]. Such regional differences may reflect a combination of genetic background, environmental exposures, and diagnostic access. Approximately 10% of ALS cases have a documented family history (fALS), while the remaining 90% are classified as sporadic (sALS). The average age at symptom onset typically ranges from 40 to 60 years for fALS and 58 to 63 years for sALS [5]. Notably, familial ALS may present as a seemingly sporadic disease due to incomplete penetrance, where individuals carrying a pathogenic mutation do not develop the disease or exhibit a delayed onset, making the familial link less obvious, or simply due to incomplete family history. While ALS heritability vary by population, ranging from 8% to 61%, this significant variability strongly supports the notion that ALS is a multifactorial disease resulting from a complex interaction of genetic, epigenetic, and environmental factors (e.g., lifestyle, diet, exercise, smoking) [6]. Intriguingly, a significant overlap in genetic risk factors and gene polymorphisms exists between sporadic and hereditary ALS, suggesting shared pathogenic mechanisms that blur the traditional distinction between these two forms and highlight the importance of investigating common underlying pathways.

The advent of advanced molecular genetic techniques, particularly long-read DNA sequencing and single-cell genomics, has revolutionized our understanding of ALS etiology, moving beyond traditional gene identification [7]. To date, over 50 genes associated with the disease have been identified, including prominent examples such as chromosome 9 open reading frame 72 (*C9orf72*), superoxide dismutase 1 (*SOD1*), the *TARDBP* gene encoding transactive response DNA-binding protein 43 (TDP-43), and the fused in sarcoma (*FUS*) gene [8]. Notably, cytoplasmic aggregates of TDP-43 proteinopathy are considered a neuropathological hallmark in nearly all ALS cases, regardless of *TARDBP* gene mutations. While coding region mutations (e.g., *SOD1, TARDBP*) explain a subset of fALS cases, non-coding alterations—including *C9orf72* repeats (accounting for~40% of fALS and 5–10% of sALS cases), regulatory variants in untranslated regions (UTRs), and splice-site modifiers—are increasingly implicated in disease mechanisms. Despite these significant discoveries, the genetic basis remains unknown for most ALS patients. This knowledge gap, combined with the newfound ability to thoroughly analyze previously inaccessible genomic regions, has spurred a growing focus on the non-coding genome, with recent studies providing compelling evidence for its pivotal role in ALS pathogenesis [9].

Emerging insights have profoundly shifted our understanding of previously overlooked genomic components, particularly transposable elements (TEs). Once controversially dismissed as ‘junk DNA,’ these mobile genetic elements are now recognized as dynamic and integral regulators of genome function, playing a critical, albeit intricate, role in shaping genetic diversity and gene expression [10]. The challenging nature of studying TE polymorphism and mobilization with conventional molecular techniques has been significantly overcome by advancements like long-read DNA sequencing, allowing for a comprehensive analysis of their genomic impact. Mounting evidence increasingly implicates dysregulated TE activity in the pathogenesis of various neurodegenerative diseases, including ALS, often contributing to disease risk and progression mechanisms (e.g., inflammation, neurodegeneration) [11]. Beyond their mutagenic potential via de novo genomic insertions and the generation of somatic mutations in post-mitotic neurons, TEs are crucially involved in a complex interplay with the epigenome. They function as both substrates for and active orchestrators of epigenetic regulation, profoundly modulating gene expression and actively reshaping the epigenetic landscape [12]. This multifaceted influence is mediated through diverse mechanisms, including alterations in DNA methylation patterns, specific histone modifications, and the transcription of various non-coding RNAs, all of which can profoundly impact neuronal function, vulnerability, and survival.

This review aims to provide a comprehensive and timely synthesis of the current understanding of TEs in ALS. While TEs were once dismissed as genomic relics, emerging evidence underscores their profound influence on gene regulation, genome integrity, and inflammatory signaling—all of which intersect with key pathological features of ALS. Recent advances in sequencing technologies and epigenetic profiling have uncovered previously hidden layers of TE involvement in neurodegeneration. Despite these insights, the field remains fragmented, with critical gaps in mechanistic understanding and therapeutic translation. By integrating current findings across genetics, epigenetics, and neuroinflammation, this review seeks to clarify the multifaceted roles of TEs in ALS pathogenesis.

## 2. Transposable Elements in the Human Genome

Transposable elements (TEs), historically considered genomic parasites, are dynamic mobile genetic elements that have profoundly shaped eukaryotic genomes. They are broadly classified into two major classes based on their distinct mechanisms of movement: Class I, retrotransposons, which mobilize via an RNA intermediate (a process termed retrotransposition), and Class II, DNA transposons, which move directly as DNA.

Retrotransposons (Class I) utilize a “copy and paste” mechanism. Their RNA transcripts are reverse transcribed into cDNA, which is then inserted into a new genomic location, thereby increasing their copy number [13]. In contrast, DNA transposons (Class II) typically move by a “cut and paste” mechanism, involving the direct excision of the DNA element from its original locus and its reintegration elsewhere in the genome without an RNA intermediate [14]. Crucially, while both Class I and Class II TEs are found in the human genome, only retrotransposons are actively mobilizing in contemporary human populations. The “cut and paste” mechanism of endogenous DNA transposons is generally considered to be inactive in humans due to the accumulation of disabling mutations over evolutionary time [15]. While some TEs encode the necessary machinery for their own movement (autonomous TEs), a significant proportion of the TE content in the human genome consists of non-autonomous elements. These non-autonomous TEs lack the full complement of genes required for mobilization and thus rely on the enzymatic machinery provided in trans by active autonomous TEs for their propagation [16] (Figure 1).

Collectively, TEs constitute a remarkable 45% of the human genome, far exceeding the protein-coding fraction. Their distribution is rarely random; instead, TEs exhibit varying degrees of insertion preference for specific genomic sites or compartments, influencing the evolution and regulation of surrounding genes [17]. Within retrotransposons, which are the focus for human disease given their ongoing activity, two main subtypes are dominant in humans: Long Terminal Repeat (LTR) retrotransposons and non-LTR retrotransposons. LTR retrotransposons are flanked by LTRs and include the abundant human endogenous retroviruses (HERVs). HERVs represent remnants of ancient retroviral infections that have become endogenized and are now stably integrated into the host germline. While generally fixed in the genome, their transcriptional activity and the production of viral-like particles can significantly contribute to genomic variation, epigenetic modulation, and host gene regulation [18] (Figure 1).

Non-LTR retrotransposons represent a diverse class of TEs that includes the most prolific active TEs in the human genome. Of these, Long Interspersed Elements 1 (LINE-1, or L1) are currently the only active autonomous non-LTR retrotransposons in the human genome. They comprise approximatively 17% of the human genome [19]. L1s encode reverse transcriptase (RT) and endonuclease (EN) enzymes which are crucial for their own retrotransposition and, notably, for mobilizing non-autonomous elements, such as Alu and SVA. Alu elements belong to the Short Interspersed Nuclear Elements (SINEs) group and are the most abundant non-autonomous retrotransposons (>10% of genomic mass). They are primate-specific and are entirely dependent on L1 protein elements for propagation. In addition, SVA (SINE-VNTR-Alu) elements represent a younger class of composite retrotransposons that are also primate-specific and L1-dependent [20] (Figure 1).

The widespread presence and dynamic activity of these elements are increasingly recognized not just as sources of genomic instability but also as critical contributors to gene regulation, chromatin organization, and phenotypic diversity. TE sequences can provide novel cis-regulatory elements, such as promoters, enhancers, and transcription factor binding sites, which can alter the expression of nearby host genes or even reshape regulatory networks [21]. Furthermore, TE activity can induce changes in chromatin accessibility and nuclear organization, influencing gene expression by acting as nucleating centers for heterochromatin formation or by shaping topologically associating domains (TADs) [22]. This intricate interplay underscores the need to comprehensively characterize TE dynamics in human health and disease.

Due to their heterogeneous mechanisms of mobilization, each subtype of active retrotransposon exhibits a distinct regulation of retrotransposition and contributes in a unique, yet profound, way to genome dynamics. The consequences of TE activity range from direct genomic alterations to subtle regulatory influences on host gene expression. For instance, the insertion of an SVA element directly into a coding gene or a critical regulatory element can drastically affect its structure or alter its expression, a phenomenon increasingly recognized in various neurological diseases. An example is a 2.6 kb SVA insertion within intron 32 of the TAF1 gene, which is causative of X-linked dystonia-parkinsonism (XDP) and significantly reduces TAF1 mRNA expression in the caudate nucleus of affected patients [23]. Similarly, a 3.1-kb SVA insertion in the Fukutin (FKTN) gene has been described as causative in Fukuyama-type congenital muscular dystrophy (FCMD) (OMIM #607440, #253800) [24] (Table 1).

Beyond these direct insertional mutations, retrotransposons are also potent architects of genomic instability, capable of triggering larger-scale chromosomal rearrangements such as deletions, duplications, inversions, and chromosomal translocations through ectopic recombination between homologous elements of the same family located at non-allelic positions [36]. This non-allelic homologous recombination (NAHR) can lead to significant structural variations with pathological consequences. Furthermore, retrotransposons can drive disease both through insertional mutations and by subtly altering host gene expression via their regulatory activity. This is mediated by their intrinsic regulatory sequences, which can act as novel promoters, enhancers, or transcription factor binding sites [37]. For example, the insertion of an endogenous retrovirus (ERV) into the C4 gene has been linked to increased C4 expression, leading to excessive synaptic pruning, a phenomenon strongly implicated in the pathogenesis of schizophrenia [38]. Moreover, the interplay between TEs and the host genome extends to sophisticated regulatory networks involving non-coding RNAs. The transcription of TEs can generate various RNA species, including long non-coding RNAs (lncRNAs) and precursors for small RNAs like microRNAs (miRNAs) and piwi-interacting RNAs (piRNAs). These TE-derived RNAs can then participate in gene silencing pathways, either by directly targeting host gene transcripts or by guiding epigenetic modifications that repress gene expression or maintain chromatin silencing. Dysregulation of these RNA-mediated TE suppression mechanisms, often observed in aging and disease, can lead to uncontrolled TE transcription and retrotransposition, contributing to cellular stress, DNA damage, and chronic inflammation, all of which are hallmarks of neurodegenerative processes [11].

The persistent presence and potential mobility of TEs in the human genome underscore a continuous evolutionary ‘arms race’ between these elements and the host’s defense mechanisms. The host employs a sophisticated repertoire of strategies to silence TEs, including widespread DNA methylation and repressive histone modifications (e.g., H3K9me3) that compact chromatin and render TEs transcriptionally inactive. Furthermore, RNA interference pathways, particularly those involving piRNAs, play a crucial role in post-transcriptional silencing and in guiding epigenetic modifications to TE loci [39]. However, TEs have evolved diverse strategies to evade these host controls, contributing to their ongoing activity. When these host suppression mechanisms are compromised, aberrant TE transcription and mobilization can occur, leading to the accumulation of TE-derived nucleic acids that can be recognized by cellular innate immune sensors, triggering an inflammatory response often characterized by interferon pathway activation, mimicking a viral infection [40]. This immunogenic aspect of TE activity is increasingly implicated in various inflammatory and autoimmune disorders, as well as neurodegenerative conditions.

Beyond their established roles in cis-regulation, TEs can also directly contribute to the coding potential of the host genome. Through a process known as exonization, TE sequences can be spliced into messenger RNA (mRNA) transcripts, giving rise to novel exons and subsequently new protein isoforms or entirely new protein-coding genes. Furthermore, TE insertions can facilitate the formation of chimeric transcripts, where sequences from a TE are fused with adjacent host gene sequences. These chimeric RNAs can encode new proteins, modify the function of existing ones, or alter mRNA stability and localization, thereby expanding the functional landscape of the human proteome and contributing to evolutionary novelty [41]. While germline retrotransposition events are significant for heritable diseases, a burgeoning field of research highlights the importance of somatic retrotransposition, where TEs mobilize within non-germline cells. This phenomenon is particularly notable in the developing brain, where L1 activity in neuronal progenitor cells can lead to genomic mosaicism within neuronal populations, potentially contributing to neural circuit diversity, but also implicated in neurodevelopmental and neuropsychiatric disorders [42]. The somatic mobilization of TEs in neurons has been demonstrated in other conditions, such as Rett syndrome [31] and schizophrenia [43], and could have similar implications in ALS, although direct evidence remains limited (Table 1). Moreover, aberrant L1 retrotransposition is frequently observed in various cancers, where new somatic insertions can act as ‘driver’ mutations by disrupting tumor suppressor genes, activating oncogenes, or altering gene expression profiles, thereby contributing to tumor initiation, progression, and heterogeneity.

The increasing understanding of TE pathogenicity has opened new avenues for therapeutic intervention. Notably, reverse transcriptase inhibitors (RTIs), which were originally developed as antiretroviral drugs for HIV, have demonstrated efficacy in suppressing L1 retrotransposition in vitro and in vivo [44]. This suggests a promising therapeutic strategy for diseases where uncontrolled TE activity is a contributing factor, including certain cancers and neurodegenerative disorders, by directly inhibiting the machinery required for their propagation. Further research is ongoing to evaluate the clinical potential and specificity of such approaches.

## 3. ALS and Transposable Elements

Mounting evidence strongly implicates dysregulated transposable element activity in the pathogenesis of ALS. While TEs are normally subjected to stringent epigenetic silencing, their de-repression has been observed in both sporadic (sALS) and familial (fALS) ALS cases, as well as in various several ALS models, including those carrying common *TDP−43* and *SOD1* mutations [45]. This aberrant TE activation is mechanistically linked to neurodegeneration through multiple pathways.

Firstly, increased TE transcription leads to the accumulation of their RNA and protein products. These products, particularly double-stranded RNA (dsRNA) derived from retrotransposons like LINE-1, Alu elements, and human endogenous retroviruses (HERVs), are recognized by the cell as foreign or viral, triggering a chronic innate immune response and neuroinflammation. This “viral mimicry” activates cytoplasmic pattern recognition receptors including the cyclic GMP-AMP synthase (cGAS)-STING pathway, a conserved immune surveillance mechanism that detects misplaced DNA (e.g., from TE retrotransposition). Alongside cGAS-STING, RIG-I-like receptors (RLRs) also respond to TE-derived dsRNA, leading to the robust production of type I interferons (IFN-I) and other pro-inflammatory cytokines (e.g., IL-6, TNF-α) [46,47]. Recent research has specifically shown that cytoplasmic synthesis of endogenous Alu complementary DNA (cDNA) can engage the cGAS pathway, leading to an inflammatory response. For example, Fukuda et al. demonstrated that Alu RNA released into the cytoplasm can be reverse-transcribed and activate cGAS-STING, contributing to age-related macular degeneration, a mechanism highly relevant to the chronic inflammation observed in ALS [48]. This sustained inflammatory signaling contributes to cellular stress, impaired proteostasis, and ultimately, neuronal damage and death.

Studies have specifically shown increased levels of HERV-K transcripts and proteins, notably the envelope (env) protein, in the brains and cerebrospinal fluid of some ALS patients. Overexpression of HERV-K or its env protein has been demonstrated to cause neurite retraction and motor neuron toxicity in in vitro and in vivo models, strikingly recapitulating aspects of ALS pathology [49]. While LINE-1, Alu, and HERVs dominate current research, emerging evidence highlights SVA (SINE-VNTR-Alu) elements as underappreciated contributors to ALS gene dysregulation. Notably, SVA_67 modulates the expression of genes in the MAPT locus (including LRRC37A4P pseudogene and MAPT), suggesting a tau-linked pathogenic mechanism [50]. Also, a polymorphic SVA upstream of FUS acts as an allele-specific enhancer, potentially explaining expression variability in ALS models.

Secondly, active TEs, particularly LINE-1 (L1) elements, can cause direct genomic instability. Through their “copy and paste” retrotransposition mechanism, new insertions can occur within genes or regulatory regions, leading to DNA damage (e.g., double-strand breaks), altered gene expression, or even insertional mutagenesis. Such genomic insults are particularly detrimental in post-mitotic motor neurons, which have limited capacity for DNA repair and regeneration, accumulating damage over time. This contributes to the concept of a ‘mosaic genome’ in neurons, where somatic TE mobilization events in individual cells can lead to cellular heterogeneity and contribute to disease progression. Furthermore, the mislocalization and aggregation of TDP-43, a key RNA-binding protein and a neuropathological hallmark in nearly all ALS cases, are intimately linked with TE dysregulation. Under physiological conditions, nuclear TDP-43 plays a crucial role in suppressing TE activity by binding to TE-derived RNA, regulating their processing and stability, and preventing their retrotransposition, thereby maintaining genome integrity [51]. When TDP-43 becomes dysfunctional due to mutation or misfolding, its nuclear depletion and cytoplasmic aggregation lead to a failure in TE suppression, allowing TEs to become de-silenced and activated. This establishes a vicious cycle where TE activation exacerbates TDP-43 pathology and vice versa, contributing to a cascade of cellular dysfunction and neurodegeneration. Research indicates that TDP-43 deficiency is associated with genome-wide epigenetic changes, including altered R-loop formation and DNA hydroxymethylation patterns at TE loci, further driving TE de-repression. This dysregulation of TE epigenetic silencing also extends to alterations in histone modifications and the machinery that controls them, contributing to a more open chromatin state that facilitates TE transcription [51]. The impact of TE activation is not limited to motor neurons; evidence suggests TE de-repression may also occur in glial cells (e.g., astrocytes and microglia), contributing to non-cell autonomous neurotoxicity and neuroinflammation.

While increased activity of LINE-1, Alu, and HERV-K elements has been repeatedly observed in ALS, their relative contribution to disease pathogenesis remains debated. Some studies emphasize the toxic effects of HERV-K env protein, which can directly induce motor neuron degeneration [29,30], whereas others highlight LINE-1–driven genomic instability and somatic mosaicism as primary drivers of neuronal dysfunction [42,52]. Alu elements, though less studied in ALS than in Alzheimer’s disease, are increasingly implicated through dsRNA accumulation and cGAS-STING activation [28,48]. These discrepancies may reflect differences in patient cohorts (sporadic vs. familial ALS), experimental models, and detection methodologies (RNA-seq vs. retrotransposition assays). A systematic comparison across models is still lacking, underscoring the need for integrative studies to clarify whether ALS pathology is predominantly shaped by one class of retroelements or by a convergent effect of multiple TE families.

The widespread presence of dysregulated TEs in the brains and spinal cords of ALS patients, particularly the activation of HERVs, LINE-1, and Alu elements, underscores their potential as significant contributors to the complex etiology of this devastating neurodegenerative disorder.

## 4. Epigenetic Regulation of Transposable Elements

To maintain genome integrity and prevent aberrant transcription or mobilization, transposable elements, particularly active retrotransposons, are subjected to a multilayered epigenetic silencing network that operates at both transcriptional and post-transcriptional levels. DNA methylation is a primary and conserved mechanism for silencing TEs, especially at promoter-containing sequences such as LTRs and LINE-1 5′ UTRs [53]. De novo methylation is catalyzed by DNA methyltransferases DNMT3A and DNMT3B, with DNMT3L acting as a regulatory cofactor in the germline, while DNMT1 maintains methylation during DNA replication [54]. DNA methylation not only represses TE transcription but also prevents the binding of transcription factors and the recruitment of RNA polymerase II to TE promoters. Specifically, studies have shown differential methylation, including global hypomethylation and hypomethylation of LINE-1 elements, in the CNS of ALS patients compared to controls, suggesting a breakdown of this key repressive mechanism. For instance, research by Pfaff et al. (2025) demonstrated increased retrotransposition-competent L1s in the genomes of individuals with ALS, directly linking epigenetic dysregulation to increased TE activity [52]. In parallel, histone modifications play an essential role in compacting TE-containing chromatin into transcriptionally inactive states. One of the most prominent repressive marks is histone H3 lysine 9 trimethylation (H3K9me3), deposited by histone methyltransferases such as SETDB1 (ESET) and SUV39H1/2 [55]. H3K9me3-marked TEs are bound by HP1 (Heterochromatin Protein 1), which promotes heterochromatin formation and restricts transcriptional access. SETDB1 is recruited to specific TE families—especially endogenous retroviruses (ERVs)—via KRAB-domain zinc finger proteins (KRAB-ZFPs), which recognize TE-derived DNA sequences and scaffold the co-repressor complex KAP1/TRIM28, along with NuRD, HDACs, and the histone chaperone DAXX/ATRX, to enforce repressive chromatin states [56]. Dysregulation of these histone modifiers and the loss of H3K9me3 have been implicated in the neurodegeneration observed in ALS, further linking epigenetic mechanisms to disease pathology. For example, studies have shown altered expression or function of H3K9 methyltransferases in ALS models and patient samples.

Another key repressive modification is histone H3 lysine 27 trimethylation (H3K27me3), catalyzed by the Polycomb repressive complex 2 (PRC2) (e.g., Enhancer of Zeste homologue 2-EZH2, Suppressor of Zeste 12—SUZ12, Embryonic Ectoderm Development—EED), which silences specific TE subfamilies during early embryogenesis and is often seen as complementary to H3K9me3, especially in stem cells [57]. Interestingly, many TEs show a bivalent chromatin state (both H3K4me3, an active mark, and H3K27me3), suggesting poised but repressed regulatory potential, particularly in developmentally regulated loci [58]. Crucially, the epigenetic silencing of TEs is not a static process but is dynamically regulated across developmental stages and exhibits considerable cell-type and tissue-specificity. While mechanisms like PRC2-mediated H3K27me3 are particularly critical for TE silencing during early embryogenesis and in pluripotent stem cells, maintaining genome integrity during differentiation, certain TE subfamilies can evade or become derepressed in specific somatic tissues, such as neurons, contributing to cell-type-specific genomic variation and potential pathological states.

Beyond transcriptional silencing, RNA-based epigenetic mechanisms provide an essential layer of post-transcriptional regulation. In germ cells and early embryos, the piRNA (PIWI-interacting RNA) pathway is critical for recognizing and degrading TE-derived transcripts and reinforcing transcriptional silencing through recruitment of DNMTs and chromatin modifiers [59]. PIWI proteins (e.g., MIWI, MILI in mice) bind piRNAs that guide them to complementary TE RNAs, triggering cleavage and co-transcriptional gene silencing. Additionally, endogenous siRNA pathways and long noncoding RNAs (lncRNAs) derived from TEs contribute to heterochromatin formation and TE repression in somatic tissues [60].

Recent evidence also highlights the involvement of chromatin remodelers such as SMARCAD1, Chromodomain helicase DNA-binding (CHD) family proteins, and SWItch/Sucrose Non-Fermentable (SWI/SNF) complexes, which help reposition nucleosomes and stabilize heterochromatin over TE loci [61]. Moreover, N6-methyladenosine (m6A) modification of TE-derived RNA has emerged as a novel layer of regulation, targeting transcripts such as LINE-1 RNA for degradation via YTHDF proteins, thereby preventing their retrotransposition [62]. Notably, SIRT6, a histone deacetylase (HDAC), has been identified as a key factor in repressing LINE-1 elements by mono-ADP-ribosylating KAP1, which facilitates heterochromatin formation [47]. SIRT6 function is known to be modulated by age and cellular stress, providing a direct link between environmental factors, aging, and TE dysregulation in neurodegeneration [47]. The multi-layered silencing network exhibits a remarkable specificity, as different TE families and even subfamilies are preferentially targeted by distinct host defense mechanisms. For instance, while Krüppel-associated box domain zinc finger proteins (KRAB-ZFPs) are pivotal for silencing many ERVs, other factors may be more critical for L1 or Alu elements, underscoring the nuanced and co-evolutionary nature of TE repression. Beyond direct molecular modifications, nuclear architecture also plays a vital role in TE silencing. Many TE loci are spatially compartmentalized into transcriptionally repressed regions within the nucleus, often localized at the nuclear periphery or around the nucleolus. Emerging evidence suggests that principles of liquid–liquid phase separation, involving key silencing factors like Heterochromatin Protein 1 (HP1), contribute to the formation of these condensed heterochromatic domains, thereby physically restricting access to the transcriptional machinery and reinforcing TE repression. Importantly, failure of any of these regulatory pathways—due to aging, oxidative stress, environmental exposure, or mutations in epigenetic regulators (e.g., DNMTs, TET2, KRAB-ZFPs)—can result in TE derepression. This leads to transcriptional noise, double-stranded DNA breaks, cytosolic DNA sensing, and activation of innate immune pathways such as cGAS-STING or RIG-I-like receptors, all of which are increasingly linked to neurodegenerative disease, dry age-related macular degeneration, and autoimmune conditions [48]. Thus, TE silencing represents a crucial genomic defense mechanism controlled by a complex, evolutionarily conserved epigenetic machinery. While the primary objective of this intricate silencing machinery is to safeguard genome integrity, it is also increasingly appreciated that the tight, yet dynamic, regulation of TEs allows for their context-dependent co-option to contribute to host gene regulation and evolutionary innovation, demonstrating a delicate balance between suppression and potential utility.

## 5. Transposable Elements as Therapeutic Targets

The understanding of TEs in ALS has rapidly advanced, paving the way for novel therapeutic strategies. This evolving comprehension, driven by recent technological breakthroughs such as long-read DNA sequencing that overcome prior challenges in analyzing transposon polymorphism, underscores the critical role of TEs in genomic instability, transcriptional dysregulation, and neuroinflammation in neurodegenerative diseases [63]. A prominent example is TPN-101 (Transposon Therapeutics), a selective LINE-1 reverse transcriptase inhibitor currently under investigation in the Phase 2/3 HEALEY ALS Platform Trial (NCT05136885). Interim results from a Phase 2 study in C9orf72-related ALS/FTD patients (NCT04993755) are highly promising, suggesting that TPN-101 may favorably impact biomarkers of neurodegeneration and inflammation, such as neurofilament light (NfL), neurofilament heavy chain (NfH), and interleukin-6 (IL-6). Furthermore, preliminary findings indicate a significant 50% reduction in the rate of decline of slow vital capacity (SVC) in the ALS group compared to placebo, suggesting a potential clinical benefit on disease progression. This Phase 2 trial has now been completed, with promising biomarker and functional outcomes, and TPN-101 has been selected for inclusion in the upcoming Phase 2/3 HEALEY ALS Platform Trial (Table 2).

Beyond direct LINE-1 inhibition, antiretroviral therapies (ARTs) traditionally used for HIV, including zidovudine and lamivudine, have also been explored to suppress Human Endogenous Retrovirus K (HERV-K) activity in ALS patients. The Lighthouse Study II (NCT06658977) was a Phase 3 clinical trial investigating the effectiveness of Triumeq in treating ALS with some reports indicating decreased HERV-K levels in patient tissues. The trial was terminated early for futility after an interim analysis showed no significant benefit on survival or functional outcomes. However, their effectiveness in producing consistent clinical benefit has been variable, likely due to limited central nervous system (CNS) penetration, insufficient specificity for retrotransposon targets, or variability in HERV-K activation among patients. This highlights the urgent need for more selective HERV-targeted approaches and a deeper understanding of HERV-mediated pathology, including the specific role of the HERV-K env protein [29] (Table 2). Despite these promising findings, several limitations of TE-targeted interventions must be acknowledged. A major challenge is the restricted central nervous system penetration of many reverse transcriptase inhibitors, which may explain the variable outcomes reported in clinical trials such as Triumeq/Lighthouse II [29]. In addition, the heterogeneity of ALS cohorts—including differences between sporadic and familial cases, or the degree of HERV-K activation—complicates patient stratification and trial design [63]. Another unresolved issue is the lack of validated biomarkers to reliably monitor TE activity and therapeutic response in vivo (Table 2). Going forward, strategies that combine direct TE inhibition with approaches targeting downstream inflammatory cascades (e.g., TE inhibition plus anti-inflammatory or epigenetic modulators such as HDAC inhibitors) may offer enhanced efficacy by tackling both the upstream source of TE activation and its pathogenic consequences [47,64]. Such combination therapies could increase the translational value of TE-based interventions in ALS.

The development of robust and accessible biomarkers for TE activity is essential for advancing TE-targeted therapies. Such biomarkers should include TE-derived transcripts and proteins, as well as TE-associated immune markers (e.g., components of the cGAS-STING pathway) in plasma or cerebrospinal fluid. These tools will be indispensable for patient stratification, monitoring disease progression, and assessing therapeutic efficacy in clinical trials.

Looking ahead, future studies must dissect the diverse mechanisms by which TEs contribute to ALS pathology. This includes evaluating the roles of other active retrotransposons such as SVA and Alu, investigating the concept of a mosaic genome arising from somatic TE mobilization in neurons, and exploring how polymorphisms within TE sequences may differentially modulate gene regulation [50]. In addition, the interplay between TEs and endogenous repressive pathways—such as those involving SIRT6, a histone deacetylase known to silence TEs—warrants closer examination [47]. While the direct efficacy of TE-targeting drugs in ALS remains under investigation, this area represents a rapidly evolving and promising therapeutic frontier, with insights increasingly drawn from TE-related mechanisms across other neurodegenerative diseases. Combination therapies targeting both TE activity and downstream inflammatory responses may offer a particularly effective strategy for disease modification.

## 6. Biochemical Parameters and Transposable Elements in ALS Patients

It is now recognized that ALS is a multi-systemic disease rather than a disorder restricted to the central nervous system. Beyond the direct neuronal degeneration, systemic clinical biochemical indicators, such as homocysteine (HCY) levels, lipid profiles, and cytokine expression, are increasingly recognized for their significant roles in disease progression and patient prognosis. Emerging evidence points to an intricate interplay between these systemic factors and the aberrant activity of transposable elements (TEs) in ALS.

Elevated HCY levels and reduced folate levels are consistently observed in the cerebrospinal fluid, plasma, and serum of ALS patients. This metabolic dysregulation is further supported by findings in *SOD1*−G93A mouse models of ALS, which exhibit reduced folate concentrations in plasma, cerebral cortex, and spinal cord [65]. HCY is a crucial intermediate in the one-carbon metabolism pathway, which is essential for the synthesis of S-adenosylmethionine (SAM). SAM serves as the universal methyl group donor for a myriad of biochemical reactions, including crucial epigenetic modifications like DNA methylation. A decrease in SAM levels, directly resulting from hyperhomocysteinemia, can lead to global DNA hypomethylation and, specifically, hypomethylation of TE loci, such as LINE-1 and endogenous retroviruses (ERVs) [66]. This epigenetic alteration contributes to their derepression and subsequent neurodegeneration in ALS. Indeed, DNA hypomethylation at TE loci, as evidenced by LINE-1 reactivation in the spinal cord of ALS patients, underscores a direct link between impaired one-carbon metabolism and TE dysregulation. This mechanism provides a clear molecular pathway through which systemic metabolic imbalances can directly influence genomic stability and gene expression in the CNS. While elevated homocysteine and reduced folate levels are consistently observed in ALS patients [67,68], the direct mechanistic link between hyperhomocysteinemia and LINE-1 activation remains only partially established. Evidence from patient tissues supports LINE-1 hypomethylation in the spinal cord of ALS cases [66], and studies in *SOD1* mouse models confirm reduced folate and SAM availability [65], suggesting a plausible pathway through impaired one-carbon metabolism. However, most studies are correlative and rely on bulk tissue analyses. Direct demonstration that hyperhomocysteinemia *causally drives* LINE-1 retrotransposition in ALS is still lacking. Future work employing locus-specific methylation profiling [66] and single-cell TE assays will be crucial to determine whether systemic one-carbon imbalance directly contributes to TE activation, or whether both phenomena reflect broader epigenetic instability in ALS.

Another important, albeit complex, aspect of ALS pathology is lipid and metabolic imbalance [69]. While several studies have reported the contribution of lipid dyshomeostasis to neurodegeneration in ALS [70], paradoxically, hyperlipidemia has also been reported as a significant prognostic factor for improved survival in ALS patients [71]. This suggests a nuanced role for lipid metabolism that is not yet fully understood. Regarding TEs, only a limited number of in vivo studies have explored the direct relationship between lipid homeostasis and TE activity in ALS models. One notable study investigated the effects of histone deacetylase (HDAC) inhibitors in a mouse model of ALS. HDAC inhibitors, known epigenetic modulators that regulate chromatin structure and gene expression, were found to stabilize lipid homeostasis and slow ALS progression [64]. This finding suggests a potential epigenetic link between lipid metabolism and TE regulation, as HDACs are critical for maintaining repressive chromatin states over TEs. Targeting HDACs could thus offer a novel therapeutic avenue to stabilize both lipid metabolism and TE-mediated pathology in ALS.

Chronic neuroinflammation is a hallmark of ALS progression, characterized by elevated levels of pro-inflammatory cytokines such as IL-1βbeta, IL-6, and tumor necrosis factor (TNF-α) observed in ALS patients [72]. Methylation profiling has further revealed specific changes in the expression of pro-inflammatory genes, particularly within T cells and monocytes, which promote a persistent inflammatory state [72]. This inflammatory milieu can directly increase TE transcription, exacerbate neuronal damage and create a vicious cycle of pathology. The contribution of endogenous retroelements to this neuroinflammation is particularly striking. For instance, a comparison between ALS patients and healthy subjects showed increased secretion levels of interleukin (IL)-6, interferon (IFN)-γ, and TNF-α in peripheral blood mononuclear cells (PBMC) after stimulation with antigenic peptides derived from HERV-K envelope (env) glycoproteins [73] (Figure 2). This provides compelling evidence that HERV-K activation directly triggers an adaptive immune response, contributing to the systemic inflammation seen in ALS.

## 7. Conclusions

The landscape of ALS research has been profoundly reshaped by the burgeoning understanding of transposable elements. Once dismissed as genomic “junk,” TEs are now unequivocally recognized as dynamic and influential genomic architects whose dysregulation contributes significantly to ALS pathogenesis. This review has highlighted the multifaceted ways in which aberrant TE activity contributes to ALS pathogenesis. Among these, LINE-1 and HERV-K elements consistently emerge as central players, collectively driving genomic instability, inflammatory signaling, and interaction with TDP-43 pathology.

Crucially, our understanding of ALS has evolved to recognize it as a multi-systemic disease, extending beyond the confines of the central nervous system. This review has underscored the critical connections between systemic biochemical dysregulations—such as hyperhomocysteinemia affecting DNA methylation and LINE-1 activity, and lipid imbalances potentially linked to epigenetic modulators like HDACs—and the aberrant activation of TEs. These systemic factors further underscore the intricate and interconnected pathways contributing to ALS pathology, blurring the traditional distinctions between familial and sporadic forms and pointing towards shared underlying pathogenic mechanisms. The inflammatory cascade, amplified by TE-derived nucleic acids and HERV-K protein products, actively contributes to neuroinflammation, reinforcing the notion of TEs as central players in the disease’s progressive nature.

The advent of advanced molecular technologies, such as long-read DNA sequencing and single-cell genomics, has been instrumental in unveiling the intricate roles of TEs in ALS. These tools allow for an unprecedented resolution in detecting somatic TE insertions and precise quantification of their transcriptional activity across different cell types within the CNS, including susceptible motor neurons and crucial glial populations. This enhanced understanding has transitioned TEs from mere bystanders to promising therapeutic targets. Initial clinical investigations, such as the ongoing trials for LINE-1 reverse transcriptase inhibitors like TPN-101 and the exploration of antiretroviral therapies, offer compelling, albeit early, evidence that modulating TE activity could be a viable strategy to slow disease progression and ameliorate neuroinflammation.

Looking ahead, several critical avenues for future research emerge. A deeper elucidation of the precise epigenetic mechanisms (e.g., specific DNA methylation patterns, histone modifications, and chromatin remodeling factors) that govern TE silencing in a cell-type-specific manner in ALS is paramount. Further investigation into the upstream triggers of TE de-repression, including the detailed impact of environmental stressors, aging, and systemic biochemical imbalances, will be crucial. The development and validation of robust, non-invasive biomarkers of TE activity in biofluids (e.g., specific TE-derived RNAs or proteins, or their associated inflammatory mediators) are essential for effective patient stratification, monitoring disease progression, and assessing therapeutic efficacy in clinical trials. Furthermore, exploring novel therapeutic modalities, such as epigenetic editing tools aimed at re-silencing specific TE families or approaches that mitigate the downstream inflammatory consequences of TE activation, holds significant promise.

Despite the comprehensive scope of this review, several limitations should be acknowledged. First, the majority of available data linking TE activity to ALS derives from preclinical models or small patient cohorts, which may limit the translational generalizability of the findings. Second, as this is a narrative review, the selection of studies may not encompass all published evidence, and the rapid pace of discovery in the field means that some very recent reports may not have been included. Furthermore, many of the associations discussed—such as those between systemic biochemical factors and TE dysregulation—remain largely correlative, and direct causal mechanisms are still incompletely understood. Looking ahead, several critical gaps remain to be addressed. The application of single-cell approaches will be essential to map TE activation in specific neuronal and glial populations, thereby resolving cell-type-specific vulnerabilities in ALS. There is also an urgent need to develop and validate non-invasive TE biomarkers in biofluids (e.g., cerebrospinal fluid, plasma, or exosomal RNA), which could enable patient stratification, disease monitoring, and therapeutic response assessment. Finally, the field would benefit from longitudinal studies integrating TE activity with systemic metabolic and inflammatory parameters to clarify whether TE dysregulation is a primary driver or a secondary consequence of ALS pathology. Addressing these gaps will be crucial for translating TE research into clinically actionable strategies.

In conclusion, the compelling and expanding evidence for dysregulated TE activity, interwoven with systemic biochemical alterations and inflammatory responses, marks a pivotal shift in our understanding of ALS. TEs represent not only a fundamental component of disease pathogenesis but also a highly attractive and novel class of therapeutic targets. Continued research in this rapidly evolving field is poised to unlock new diagnostic tools and effective treatments, ultimately bringing us closer to overcoming the devastating challenges posed by ALS.

## Figures and Tables

**Figure 1 pharmaceuticals-18-01573-f001:**
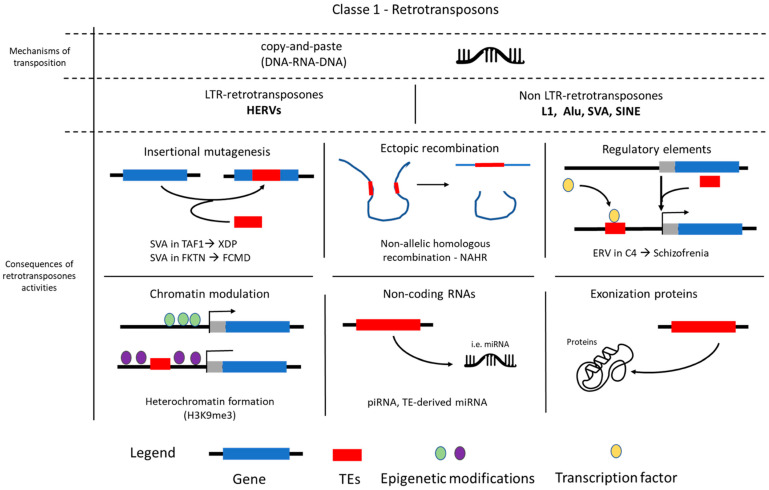
Transposable elements classification, mechanisms and impact on genome. Schematic representation showing the classification of transposable elements, mechanisms of transposition, and examples of how TEs can influence genomes.

**Figure 2 pharmaceuticals-18-01573-f002:**
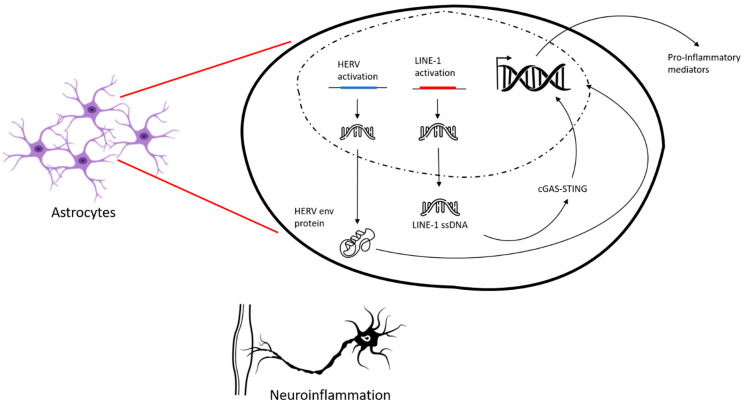
Activation of neuroinflammatory inflammation through the activation of endogenous retrotransposable elements. In pathological conditions, the endogenous retroviruses HERV (blue) and LINE-1 (red) can be activated. Activation of HERV leads to the production of the envelope protein (HERV env protein), while activation of LINE-1 produces single-stranded DNA (LINE-1 ssDNA).

**Table 1 pharmaceuticals-18-01573-t001:** Transposable Elements Implicated in Human Diseases.

Associated Disorders	TEs Type	TEs Activity	References
AD	Alu LINEs	Increased Alu RNA processing in the brain → altered gene expression (e.g., genes related to synaptic transmission and inflammation). Accumulation of cytoplasmic Alu RNA correlates with oxidative stress and neuronal damage. Widespread activation of L1 and ERVs associated with neurofibrillary tangle burden.	[25,26]
HD	LINEs	Inverse correlation between L1 expression and CAG repeat length	[27]
ALS/FTD	Alu	Dysregulation of transcripts of Alu elements including AluYk12 and AluYa5	[28]
ALS	HERVs LINEs	HERV-K potentially contributes to neurodegeneration due to toxic effects of HML-2 Envelope (Env) protein; Overexpression of the HERV-K(HML-2) envelope protein causes motor neuron toxicity and motor dysfunction in transgenic mice; L1 retrotransposition	[29,30]
Rett syndrome	LINEs, Alu	Methylated MeCP2 is involved in the control of L1 mobility in the nervous system; MeCP2 represses L1 expression and transposition; In MECP2-KO mouse models, L1 is derepressed and can transpose into the brain.	[31]
MS	HERVs	Increased expression of HERV-H, HERV-W families.	[32]
Schizophrenia	HERVs	Higher HERV-W env RNA expression	[33]
Ataxia telangiectasia	LINEs	L1 retrotransposition	[34]
XDP	SVA	Reduced TAF1 expression due to SVA element integration Hypermethylation of the TAF1 SVA insertion	[35]

AD: Alzheimer’s disease; HD: Huntington’s disease; ALS/FTD: Amyotrophic Lateral Sclerosis/Frontotemporal dementia; XDP: X-linked dystonia-parkinsonism; MS: multiple sclerosis; HERVs: human endogenous retroviruses; SVAs: SINE-VNTR-Alu elements.

**Table 2 pharmaceuticals-18-01573-t002:** Current and Investigational Therapies Targeting Transposable Elements in ALS.

Therapy/Strategy	Targeted TE	Mechanism of Action	Clinical Trial	Phase	Notes/Biomarkers
TPN-101 (Transposon Tx)	LINE-1	Selective reverse transcriptase inhibition	NCT05136885 (HEALEY)	Phase 2/3	↓ NfL/NfH, ↓ IL-6, 50% slower SVC decline
TPN-101	LINE-1	Same mechanism, tested in *C9orf72* ALS/FTD	NCT04993755	Phase 2	Completed. Promising biomarker effects; initial ALS-specific results encouraging
Triumeq (ART: ABC therapy)	HERV-K	Inhibits retroviral replication (RT inhibition + integrase block)	NCT06658977 (Lighthouse II)	Phase 3	Terminated early (futility: inconsistent efficacy despite reduction of HERV-K env levels
Zidovudine (AZT)	HERV-K	Nucleoside RT inhibitor (legacy HIV drug)	Preclinical	N/A	↓ HERV-K expression in vitro; limited CNS penetration
Lamivudine	LINE-1, HERVs	Broad RT inhibition	NCT03280198	Phase 2	TE suppression in vitro; no published results
HDAC inhibitors	Indirect (LINE-1/HERVs)	Epigenetic silencing via chromatin remodeling	NCT05021536 (PHOENIX trial)	Phase 3	↓ neuroinflammation, improved epigenetic repression

## Data Availability

No new data were created or analyzed in this study. Data sharing is not applicable to this article.

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
