# Peer review of "The Other Side of the Same Coin: Beyond the Coding Region in Amyotrophic Lateral Sclerosis"

_pharmaceuticals, 2025, doi:10.3390/ph18101573_

Round 1
Reviewer 1 Report
Comments and Suggestions for Authors
Dear Authors
"The other side of the same coin: beyond the coding region in ALS" is the title of the manuscript pharmaceuticals-3911095, which I reviewed and believe should be published in this journal after the following minor corrections:
Title: Add full form of ALS in the title.
Introduction:
- Include continent or country-specific epidemiological/prevalence data.
- Update the following information with recent references.
“To date (2016?), over 40 genes associated with the disease, including prominent examples such as chromosome 9 open reading frame 72 (C9orf72), superoxide dismutase 1 (SOD1), the TARDBP gene encoding transactive response DNA-binding protein 43 (TDP-43), and the fused in sarcoma (FUS) gene”.
- Add a Table, after Para 2 to highlight the associated genes and encoding proteins
Add a separate heading for the review article's limitations.
References must be corrected in accordance with journal formatting.
Author Response
Comments and Suggestions for Authors
Dear Authors
"The other side of the same coin: beyond the coding region in ALS" is the title of the manuscript pharmaceuticals-3911095, which I reviewed and believe should be published in this journal after the following minor corrections:
Title: Add full form of ALS in the title.
- Thanks to the reviewer's suggestion, we've modified the title to include the full name of ALS.
Introduction:
- Include continent or country-specific epidemiological/prevalence data.
- We thank the reviewer for this suggestion. In the Introduction (page 2, lines 56-62) , we have added updated continent- and country-specific epidemiological data, highlighting differences between Europe, North America, Asia, and other regions.
- Update the following information with recent references.
“To date (2016?), over 40 genes associated with the disease, including prominent examples such as chromosome 9 open reading frame 72 (C9orf72), superoxide dismutase 1 (SOD1), the TARDBP gene encoding transactive response DNA-binding protein 43 (TDP-43), and the fused in sarcoma (FUS) gene”.
- We appreciate the reviewer’s comment. In the Introduction (page 2, line 81), we have modified the figure by replacing the number 40 with a more up-to-date reference (2024).
- Add a Table, after Para 2 to highlight the associated genes and encoding proteins
- We thank the reviewer for this constructive suggestion. We fully agree that the genetic landscape of ALS is fundamental for understanding disease mechanisms. However, the focus of our review is specifically on non-coding regions and transposable elements (TEs) in ALS, rather than a comprehensive overview of coding mutations and associated proteins, which has been extensively covered in other recent reviews. Adding a detailed gene–protein table here could risk shifting the scope away from our main theme and duplicating information already available in the literature. We hope the reviewer agrees that this approach preserves the clarity and scope of the review while ensuring readers have access to the relevant genetic background through the cited references.
Add a separate heading for the review article's limitations.
- We thank the reviewer for this suggestion and fully agree on the importance of clearly acknowledging the limitations of our work. Rather than creating a stand-alone section, we have chosen to integrate these considerations directly into the Conclusion (pages 16-17, lines 628-659), where we discuss both the current limitations of the field and the critical gaps that remain to be addressed. In our view, this integrated format avoids redundancy and provides a more coherent narrative that connects the review’s limitations with future research directions.
References must be corrected in accordance with journal formatting.
- We are thankful to the reviewer for their valuable feedback. References have been edited in accordance with journal guidelines.

Reviewer 2 Report
Comments and Suggestions for Authors
This review manuscript addresses an emerging and highly relevant aspect of ALS pathogenesis: the contribution of transposable elements (TEs) and other non-coding genomic regions. The authors provide a comprehensive and updated synthesis of recent findings, integrating molecular mechanisms, epigenetic regulation, systemic biochemical factors, and translational therapeutic approaches. The manuscript is well-organized, clearly written, and appropriately referenced. It has the potential to make a valuable contribution to the literature on ALS. However, the review occasionally leans toward summarization rather than offering a critical evaluation of controversies, limitations, and knowledge gaps. Strengthening this analytical perspective would substantially improve the manuscript.
Recommendations for Authors:
1. The review provides a broad overview, but often presents findings without sufficient evaluation of conflicting results. For example, while the role of LINE-1, Alu, and HERVs is emphasized, the relative contributions and controversies in different ALS models and patient cohorts should be more explicitly addressed.
2. The section on therapeutic interventions (e.g., TPN-101, Triumeq, lamivudine, HDAC inhibitors) is informative, but it would benefit from a deeper discussion of limitations such as CNS penetration, patient stratification, and biomarker validation. Including a perspective on combination therapies (TE inhibition + anti-inflammatory approaches) would strengthen translational value.
3. Figures are detailed but complex. Adding a simplified model that summarizes the interplay between TE activation, epigenetic dysregulation, inflammation, and ALS progression would improve clarity. Similarly, tables (e.g., Table 1 and Table 2) should be standardized for formatting consistency.
4. The discussion of systemic factors (e.g., homocysteine metabolism, lipid profiles) is highly valuable, but the link to TE dysregulation could be more critically elaborated. For example, how robust is the evidence connecting hyperhomocysteinemia to LINE-1 activation in ALS?
5. The conclusion highlights promising areas, but it could be expanded to identify specific gaps (e.g., the need for single-cell TE profiling in ALS patient tissues, or the development of non-invasive TE biomarkers in biofluids).
6. The abstract is informative but should include a clear take-home message regarding the translational implications of TE research in ALS.
7. Some redundancies exist (e.g., repeated emphasis on LINE-1 and HERV-K). A more concise synthesis would enhance readability.
8. Minor typographical and formatting inconsistencies are present, particularly in references and table layout.
9. Please clarify the status of clinical trials mentioned (e.g., TPN-101 Phase 2/3, Lighthouse II). Indicating whether they are ongoing, completed, or discontinued would be helpful for readers.
Comments on the Quality of English LanguageThe English could be improved to more clearly express the research.
Author Response
This review manuscript addresses an emerging and highly relevant aspect of ALS pathogenesis: the contribution of transposable elements (TEs) and other non-coding genomic regions. The authors provide a comprehensive and updated synthesis of recent findings, integrating molecular mechanisms, epigenetic regulation, systemic biochemical factors, and translational therapeutic approaches. The manuscript is well-organized, clearly written, and appropriately referenced. It has the potential to make a valuable contribution to the literature on ALS. However, the review occasionally leans toward summarization rather than offering a critical evaluation of controversies, limitations, and knowledge gaps. Strengthening this analytical perspective would substantially improve the manuscript.
Recommendations for Authors:
- The review provides a broad overview, but often presents findings without sufficient evaluation of conflicting results. For example, while the role of LINE-1, Alu, and HERVs is emphasized, the relative contributions and controversies in different ALS models and patient cohorts should be more explicitly addressed.
- We appreciate this observation. We have now expanded Section 3 (“ALS and Transposable Elements”, page 9, lines 339-349) to more explicitly highlight controversies across ALS models and patient cohorts. In particular, we compare studies that report strong HERV-K env protein toxicity with others emphasizing LINE-1–driven genome instability, noting differences in methodology and sample sources. We also added the discussion on the underexplored role of SVA retrotransposons.
- The section on therapeutic interventions (e.g., TPN-101, Triumeq, lamivudine, HDAC inhibitors) is informative, but it would benefit from a deeper discussion of limitations such as CNS penetration, patient stratification, and biomarker validation. Including a perspective on combination therapies (TE inhibition + anti-inflammatory approaches) would strengthen translational value.
- We thank the reviewer for this comment. In Section 5 (page 12, lines 473-486), we now include a dedicated paragraph discussing key limitations of TE-targeted therapies, including CNS penetration, patient stratification, and biomarker validation. We also added a perspective on combination therapies (e.g., TE inhibition plus anti-inflammatory approaches), which may enhance translational potential.
- Figures are detailed but complex. Adding a simplified model that summarizes the interplay between TE activation, epigenetic dysregulation, inflammation, and ALS progression would improve clarity. Similarly, tables (e.g., Table 1 and Table 2) should be standardized for formatting consistency.
- We appreciate the reviewer’s insightful comment. A simplified schematic integrating TE activation, epigenetic dysregulation, inflammation, and ALS progression is already provided in the Graphical Abstract, which we believe effectively fulfills this purpose. To avoid redundancy, we have not included an additional version in the main figures. In line with the reviewer’s suggestion, we have carefully revised and standardized the formatting of Table 1 and Table 2 to ensure clarity and consistency.
- The discussion of systemic factors (e.g., homocysteine metabolism, lipid profiles) is highly valuable, but the link to TE dysregulation could be more critically elaborated. For example, how robust is the evidence connecting hyperhomocysteinemia to LINE-1 activation in ALS?
- We added new discussion in Section 6 (pages 14, lines 536-546) clarifying the mechanistic evidence that hyperhomocysteinemia induces LINE-1 hypomethylation in ALS. We also note that the evidence linking lipid profiles to TE activity remains preliminary, requiring further validation.
- The conclusion highlights promising areas, but it could be expanded to identify specific gaps(e.g., the need for single-cell TE profiling in ALS patient tissues, or the development of non-invasive TE biomarkers in biofluids).
- We thank the reviewer for this insightful suggestion. In the Conclusion (pages 16-17, lines 628-659), we have expanded the discussion to explicitly highlight key research gaps, including: the need for single-cell TE profiling in ALS patient tissues, the development of non-invasive TE biomarkers in biofluids for patient monitoring, and the importance of longitudinal studies integrating systemic factors with TE activity. These additions clarify directions for future investigation and emphasize translational opportunities.
- The abstract is informative but should include a clear take-home message regarding the translational implications of TE research in ALS.
- We thank the reviewer for this suggestion. We have revised the Abstract to include a clear take-home message emphasizing the translational implications of TE research in ALS, particularly the potential of TEs as biomarkers and as therapeutic targets for disease modification.
- Some redundancies exist (e.g., repeated emphasis on LINE-1 and HERV-K). A more concise synthesis would enhance readability.
- We agree with the reviewer. To reduce redundancy, we have streamlined the Conclusion (Section 7, page 15, lines 590-594) by condensing the repeated description of LINE-1 and HERV-K mechanisms into a concise synthesis. The detailed mechanistic discussion is retained in Section 3, while the Conclusion now provides a succinct integrative summary.
- Minor typographical and formatting inconsistencies are present, particularly in references and table layout.
- We thank the reviewer for this comment. We have carefully revised the manuscript to correct typographical inconsistencies and standardized the formatting of the references, tables, and abbreviations. Tables 1 and 2 now follow a uniform structure, and reference formatting has been harmonized across the manuscript.
- Please clarify the status of clinical trials mentioned (e.g., TPN-101 Phase 2/3, Lighthouse II). Indicating whether they are ongoing, completed, or discontinued would be helpful for readers.
- We thank the reviewer for this suggestion. We have updated Section 5 (page 11-12, lines 458-460 and 466-468) and Table 2 to clearly indicate the status of the clinical trials discussed. These clarifications will provide readers with a clearer understanding of the translational landscape.

Round 2
Reviewer 2 Report
Comments and Suggestions for Authors
The article has been revised in accordance with the previous comments, and in my opinion, it can be published in its current version.